# Modeling of Magnetic Scaffolds as Drug Delivery Platforms for Tissue Engineering and Cancer Therapy

**DOI:** 10.3390/bioengineering11060573

**Published:** 2024-06-06

**Authors:** Matteo B. Lodi, Eleonora M. A. Corda, Francesco Desogus, Alessandro Fanti, Giuseppe Mazzarella

**Affiliations:** 1Department of Electrical and Electronic Engineering, University of Cagliari, via Marengo 2, 09123 Cagliari, Italy; matteob.lodi@unica.it (M.B.L.); eleonorama.corda@unica.it (E.M.A.C.); mazzarella@unica.it (G.M.); 2Consorzio Nazionale Interuniversitario per le Telecomunicazioni (CNIT), Cagliari Research Unit, Department of Eletrical and Electronic Engineering, University of Cagliari, via Marengo 2, 09123 Cagliari, Italy; 3Department of Mechanical, Chemical and Material Engineering, University of Cagliari, via Marengo 2, 09123 Cagliari, Italy; francesco.desogus@unica.it

**Keywords:** cancer therapy, drug delivery, electromagnetic fields, magnetic nanoparticles, magnetic scaffolds, tissue engineering

## Abstract

Magnetic scaffolds (MagSs) are magneto-responsive devices obtained by the combination of traditional biomaterials (e.g., polymers, bioceramics, and bioglasses) and magnetic nanoparticles. This work analyzes the literature about MagSs used as drug delivery systems for tissue repair and cancer treatment. These devices can be used as innovative drugs and/or biomolecules delivery systems. Through the application of a static or dynamic stimulus, MagSs can trigger drug release in a controlled and remote way. However, most of MagSs used as drug delivery systems are not optimized and properly modeled, causing a local inhomogeneous distribution of the drug’s concentration and burst release. Few physical–mathematical models have been presented to study and analyze different MagSs, with the lack of a systematic vision. In this work, we propose a modeling framework. We modeled the experimental data of drug release from different MagSs, under various magnetic field types, taken from the literature. The data were fitted to a modified Gompertz equation and to the Korsmeyer–Peppas model (KPM). The correlation coefficient ($R^{2}$) and the root mean square error (RMSE) were the figures of merit used to evaluate the fitting quality. It has been found that the Gompertz model can fit most of the drug delivery cases, with an average RMSE below 0.01 and $R^{2}>0.9$. This quantitative interpretation of existing experimental data can foster the design and use of MagSs for drug delivery applications.

## 1. Introduction

Tissue engineering (TE) and cancer therapeutics (CThs) have been enabled by the development of biocompatible tissue-mimicking biomaterials (e.g., metals, biopolymers, and bioceramics). Bioactive materials, manufactured as tissue scaffolds, are designed to elicit specific biological responses, which are crucial for controlled healing and regeneration or therapy [1,2,3]. For 3D scaffolds, a sufficient porosity (50–80%, depending on the tissue site), as well as pore size distribution (1 μm–250 μm), must be guaranteed to ensure tissue growth, adequate biomolecule signaling, cellular homing, and vascularization [4,5]. Kim et al. [6] developed a porous polycaprolactone (PCL) scaffold for bone tissue incorporating cuttlefish bone-derived hydroxyapatite (Hap) powder to demonstrate that, in vitro, the porosity influences the proliferation and differentiation by creating an adequate biomechanical microenvironment for tissue regeneration. However, some strategies for scaffold designs do not meet TE goals [7,8,9]. Therefore, new solutions and alternative strategies to control cell–biomaterial interactions have been considered. 

Biomaterial scaffolds have been explored as devices and platforms for controlled drug delivery (DD) aimed at manipulating tissue repair and/or therapeutic outcomes [10,11]. Several physical methods in a biomaterial to exert a therapeutic action are available [12,13]. For TE applications, biomaterials for DD have the aim of providing growth factors (GFs) around the implant region to control and manipulate tissue repair, acting on cell migration, proliferation, differentiation, or, for cancer therapy (CT), exerting an anti-proliferative action [2,8,12,14]. Initially, scaffold DD systems for TE relied on mechanisms, such as molecular diffusion, material degradation, or cell migration, which are poorly controllable [14]. This new approach allows us to recreate and mimic the in vivo release profiles of factors produced during natural tissue morphogenesis or repair. For instance, GFs, such as platelet-derived growth factor (PDGF) embedded in implanted polymeric (e.g., PLGA and PLA) formulation or alginate hydrogels, were used for endothelial cell proliferation with applications in angiogenesis and wound healing [15]. Bone morphogenetic protein-2 (BMP-2) when combined with hydrogels or PLGA scaffolds is useful for modulating cell proliferation and tuning bone regeneration [15]. The release of drugs and GFs from biomaterials is not exempt from shortcomings and limitations. Indeed, not all biomaterials with local DD exhibit a spatial and temporal controlled release and a sustained drug release behavior to ensure an optimum controlled therapy, thus avoiding side effects [16,17,18]. 

Therefore, bioengineers proposed to trigger and/or regulate the delivery of biological agents (e.g., drugs and cells) using external cues and physical stimuli, thus overcoming traditional DD limitations [19]. Potential candidates as therapeutic scaffolds used in DD applications, such as TE and CT, are called stimuli-responsive scaffolds [19,20]. Stimuli-responsive scaffolds are smart biomaterial implants that can respond to exogenous or endogenous physical and/or chemical changes [19,20,21]. Several active biomaterials, responsive to external stimuli, were proposed in the literature, such as the temperature-responsive injectable hydrogel scaffold [12] and pH-sensitive scaffold [10]. Furthermore, various physical fields and energy forms (e.g., mechanical, electric, piezoelectricity, etc.) were analyzed for controlled delivery with improved safety and efficiency, while enabling new therapies [22,23,24,25]. Despite the disruptive potential of stimuli-responsive biomaterials, some limitations and challenges must be underlined. Indeed, forms of energy, such mechanical, thermal, and ultrasound energy, are not specific, reach limited penetration depths, or, instead, lead to complex technological implementations [19,20,21,22,23,24,25,26].

In the framework of stimuli-responsive scaffolds, electromagnetic (EM) energy can play a key pivotal role, overcoming DD limitations. Indeed, the EM spectrum, especially ranging from very low frequencies (i.e., few Hz) to radiofrequency (herein, hundreds of kHz), can be used to control the response of scaffolds and trigger specific effects and actions on cells and tissues for both TE and CT remotely, noninvasively, and precisely [27,28,29]. Electric field-responsive scaffolds have been proposed [29], but they cannot be easy to reach if implanted in deep body sites. On the other hand, magnetic fields (MFs) are preferred for some biomedical applications since they have a higher penetration depth and high specificity. Therefore, the possibility of manufacturing a biomaterial able to respond to the magnetic field was investigated, too [30,31,32,33]. A magnetic implant can be achieved by incorporating specialized magnetic biomaterials in a nano-formulation into the biomaterial matrix (see Figure 1), thus conferring magnetic properties to the structure, which can then be controlled spatiotemporally and remotely [30,31,32,33]. 

Magnetic nanoparticles (MNPs) are particles (<200 nm in size) composed of magnetic elements, such as iron (Fe), nickel (Ni), cobalt (Co), or their oxides (e.g., magnetite, maghemite, etc.) [34]. Zn- and Mn-substituted magnetite MNPs hold therapeutic potential against colorectal cancers [35,36]. If MNPs are embedded in biomaterials such as bio-ceramics or biopolymers, thus creating a so-called magnetic scaffold (MagS), theragnostic and multifunctional abilities are provided to scaffolds, creating new usage and applications. By varying the magnetic field strength in space or time, it is possible to control the physical, structural, and mechanical properties of these magneto-responsive scaffolds. Therefore, MagSs can be used for TE, DD, or CT [37]. MagSs can be activated (i) by static or very low-frequency MF-triggering mechanical forces and deformations, or (ii) by alternate MFs for magnetothermal conversion. 

In regard to TE, MagSs act as mechano-transducers modifying local Ca^2+^ fluxes. In [38], magnetic Hap scaffolds were cultured in vitro with pre-osteoblast and osteoblast cells (i.e., ROS 17/2.8 and MC3T3-E1, respectively) with and without an exterior static MF (~15 kA/m), finding that proliferation and differentiation were influenced. MagSs have been evaluated for cardiac tissue: a functional cardiac patch of microporous alginate scaffold impregnated with MNPs and the application of a 5 Hz external MF stimulation has been studied in [39]. The study of PCL/gelatin 3D magnetic nanofibrous constructs comprising MNPs has been carried out in [40]. 

On the other hand, MagSs can be used as therapeutic agents by exploiting the highly efficient magnetothermal conversion that MNPs embedded in a biomaterial matrix experience, if an RF MF is applied [37]. The dissipated heat can be exploited to administer hyperthermia at local and interstitial levels against solid cancers, such as bone or ductal tumors [37]. 

The intrinsic multifunctional nature of MagSs, in particular, the mechano-transducer and magnetothermal conversion features, have been exploited to implement an innovative DD platform for GFs or drug administration, as shown in Figure 1, as epitomized by the magnetic sponge loaded with docetaxel (DTX), whose release is triggered by static MF-induced (~50–350 mT) reversible mechanical deformations [41]. Exploiting a similar mechanism, in [42], hollow-fiber alginate/iron oxide nanoparticle scaffolds were prepared by 3D printing, and the MF-mediated delivery of encapsulated drugs (e.g., doxorubicin—DOX), protein, and mesenchymal stem cells was tested in vitro and in vivo. On the other hand, in [43,44], composite ethylcellulose membrane scaffolds with embedded thermosensitive poly(n-isopropyl acrylamide) (polyNIPAm)-based nanogels and MNPs exposed to 220−260 kHz, 0−20 mT MF proved to be able to increase membrane permeability as the dissipated magnetic heating increased the membrane temperature.

Recently, in [44], we dealt for the first time with the mathematical modeling of the magnetic drug delivery of growth factors to evaluate the effectiveness of MagSs as an in situ attraction platform for MNPs carrying GFs to control the bone regeneration process. The proposed DD strategy is a combination of different administration strategies mediated by different MF types (see Figure 1). Indeed, simulations to evaluate how a static MF can be used to force and drive MNPs+GFs to the MagSs were performed. Then, the in silico study of how RF MFs can be used to trigger GF release lead to the findings that the quality of regenerated bone tissue can be improved using MagSs. 

From this introductory discussion, it is possible to infer that MagSs have high potential for DD. Indeed, MagSs can overcome the significant common problems for traditional biomaterials used for DD applications, such as burst release, heterogeneity in the release phase of the bioactive agent, inhomogeneous spatial distribution, control long-term release, reduce the leakage of drugs or GFs, avoiding side effects, or the impossibility of re-loading the biomaterial [10,11,25,44]. However, from the above discussion, it can be observed clearly that there are several types of MagSs, while presenting different and fuzzy features, as well as being characterized by various DD mechanisms. This work is motivated by the need for an engineering and quantitative rationale that can drive and lead their design and use for TE and DD applications. Traditionally, DD and biomaterials for DD find strong bases in mathematical modeling and kinetics models. For MagSs, it must be noted and highlighted that very few or no models were developed to interpret their response as DD platforms. Therefore, in this work, for the first time, we focus on the physical and mathematical modeling of MagSs as innovative structures for delivering bioactive agents, in the pursuit of achieving targeted, prolonged, and stimulus-responsive release. The aim is to provide a solid framework to empower and further develop the MagS design and DD applications. To this aim, in Section 2, we performed a literature analysis to select the most relevant cases study of MagSs used as DD platforms. Then, as explained in Section 3, the experimental data from DD experiments were digitized and fitted to kinetic models. Then, in Section 4, the results are presented, and the release and kinetic parameters are linked and analyzed with respect to the intrinsic magnetic features of MagSs, and an extensive critical discussion is provided too. In Section 5, the conclusions are reported. 

## 2. Related Works and Cases Study

### 2.1. Methodology for the Literature Analysis

A literature search aimed at identifying all relevant articles was based on the selection of works to identify some cases studies of MagS DD applications. The search strategy, including all identified keywords, index terms, and abstract, has been adapted for each included database and/or information source. Studies published in the English language from January 2009 to 2024 were included. The databases used in the research included Wiley, National Institutes of Health (NIH), Scopus, PubMed, Science Direct, and IEEEXplore. We focused our attention on different aspects, namely, the biomaterial matrix; the type of MNPs (e.g., magnetic features and size); MagS manufacturing; the type of DD strategies triggered, modulated, and controlled by an external magnetic field (e.g., static, dynamic, magneto-thermal conversion, etc.); and, finally, if experimental tests were performed and for which DD applications they were proposed. In this respect, we critically and thoroughly analyzed these literature sources and carefully identified the knowledge gaps to propose a quantitative framework to study MagSs for DD. 

### 2.2. Literature Analysis

The conducted literature research led to the identification of some different, specific articles for MagSs and DD applications [41,42,43,44,45,46,47,48,49,50,51,52,53]. The results of our literature analysis are reported in Table 1. 

The selected articles offer a comprehensive overview of the different methodologies and approaches used in MagSs for targeted DD, especially in the context of TE and CT. From works [41,42,43,44,45,46,47,48,49,50,51,52,53], as can be seen from Table 1, the preferred biomaterial matrix formulation, which is a fundamental factor for achieving mechanical and biocompatibility properties, is polymeric, allowing the easy manufacturing of magnetic nanocomposite and drug- or biomolecule-loaded scaffolds. 

We hypothesize that, for MagSs, the selection of the magnetic nanoparticles to embed or the magnetic phase to synthesize is crucial. MNPs can have a different magnetism. They can be ferrimagnetic or ferromagnetic (for diameters in the range of 25–50 nm to 100 nm), i.e., they can be intrinsically magnetic and possess a permanent magnetic moment [34,45]. On the other hand, MNPs (diameter below 25 nm) can respond to an externally applied magnetic field being superparamagnetic (SPM). In any case, MNPs’ magnetism play an important role in enhancing DD efficiency. In Table 1, we can see that ferromagnetic and SPM particles are used. Ferromagnetic particles have been used in [41]. On the other hand, SPM MNPs are preferred. For instance, in [46], a 3D-printed mesoporous bioactive glass (MBG)-PCL scaffold with SPM magnetite nanoparticles was proposed for TE and DD for CT applications. MBG possesses a more optimal surface area, nanopore volume, controlled drug delivery properties, and in vivo biocompatibility, and this makes the structure more suitable and effective for the specific applications studied. The particularity of Fe_3_O_4_/MBG/PCL composite scaffolds has been accentuated by the MNPs’ presence that made these structures able to respond to the external magnetic field. The release of 20 mL of DOX was evaluated after the application of an alternating MF of 18 mT with an amplitude of 409 kHz for 30 min. In [47], the formulation of multilayer magnetic gelatin membrane scaffolds blended with Fe_3_O_4_ SPM MNPs was proposed. Gelatin MagSs are supposed to be used as in situ attraction sites for magnetized DD agents carrying GFs or drugs. However, SPM MNPs can also be used to perform DD based on the magneto-thermal mechanism. Chemical routes for doping bio-ceramics and producing in situ MNPs lead to an interesting sub-class of MagSs. For instance, in [48], magnetic hybrid composites made of (Fe^2+^/Fe^3+^)-doped Hap nanocrystals nucleated on self-assembling collagen fibers were prepared using a biologically inspired mineralization process. DOX was adsorbed onto Hap and released through the application of pulsed electromagnetic fields (PEMFs). These MagSs were tested as DD agents against osteosarcoma cancers [47,48]. In [49], a magnetic mesoporous glass formulation for a Fe_3_O_4_/CaO/SiO_2_/P_2_O_5_ system is proposed. A dynamic MF with a strength value of 1.47 kA/m and frequency of 232 kHz was used to trigger, via magneto-thermal conversion, the in vitro release of 20 ml of gentamicin, thus proving the potential of this MagS for the regeneration of a critical-size bone defect [50]. The versatility of MagSs as DD platforms is limitless. Indeed, in [51], a macroporous ferrogel is manufactured by incorporating ferrite SPM MNPs and mitoxantrone (300 g); plasmid DNA and chemokines (SDF 1-α) were released under the action of a dynamic MF (38 A/m, 120 cycles (on/off), for 2 min). In this framework, new studies dealing with MagSs as DD systems are being published [52,53,54,55] and the interest of the scientific community is very high in this topic.

### 2.3. Knowledge Gaps and Goals

From the above discussion and from Table 1, we can underline that key differences exist in MagS manufacturing. Depending on the combination of the biomaterial, MNPs and considering the intended DD applications, various mechanism of drug release are possible. 

As can be observed in Table 1, several combinations of MNPs and biomaterials, different manufacturing approaches have been proposed, but the role of formulation in DD has been poorly investigated. A rationale or set of rules for driving the selection of biomaterials and MNPs is missing, as well as to identify the best manufacturing approach. This is further complicated by the fact that the magnetic response of MagSs, given that the MNPs interact with the complex material structure, cannot be easily interpreted a priori. Furthermore, the different release mechanisms obey different physical laws, where the MagSs’ magnetic features play a pivotal role that has been poorly modeled and understood, to date. Therefore, despite the fact that the production techniques of these nano-systems have been carefully studied and tested with a proof of concept to test the release, there is a lack of theoretical or computational models to study, interpret, and design MagSs. The proposal and verification of such models are necessary to deal with MagS designs, treatment planning, and the investigation of the biological effects. In this work, we identified the difficulty to find an appropriate model suitable for modeling MagSs as DD agents. However, mathematical modeling has been widely employed in pharmacokinetics and DD, also for magnetic nano-formulations, such as in [56,57]. Furthermore, very few MagS-related studies reported the drug concentration profile as a function of release time [41,42,43,44,45,46,47,48,49,50,51,52,53]. Thus, in this work, for the first time, we will focus on works [41,43,44,46,48,49,50,51], since suitable, clear, and exhaustive experimental data for testing MagSs as DD platforms for TE and CT have been provided. These data have been studied and used to identify suitable models to apply the results to the understanding of the physical phenomena, mechanisms, and formulations underlying the interaction between the MF and MagSs for DD. 

## 3. Model and Methodology

### 3.1. Data Retrieval

The data from studies [41,43,44,46,48,49,50,51] were digitized using the online software “PlotDigitizer” [58]. We retrieved the concentration of released drug for each time for all the data found in these literature references. For pre-processing, the data were normalized. The post-processing of the obtained data and subsequent comparison of the various candidate models were performed using the “Curve Fitting Toolbox” from MATLAB 2023a (The MathWorks Inc., Boston, MA, USA). 

### 3.2. Modeling

A remark is in order. The aim of this work is the modeling of MagSs as DD agents, linking drug release to the EM properties. To this aim, the available models from pharmacokinetics and DD were used. Therefore, in the following section, we will describe the well-known pharmacokinetic models used to describe the mechanisms governing the release of bioactive agents. A comprehensive overview of DD models is provided in [59,60,61,62,63]. Given the availability of different models, several fittings were performed to identify the kinetic model that best fitted the studied data to describe the DD modalities from several MagSs [41,43,44,46,48,49,50,51] based on the available experimental data. In this work, we will focus on Gompertz and Korsmeyer–Peppas (KPM) models, proving that they are flexible and generally applicable [59,60,61,62,63]. 

#### 3.2.1. Gompertz Model

A modified Gompertz model allows studying the dissolution profile of a pharmaceutical dosage [59,60,61]:(1)$$X\left(t\right)=e^{-\alpha e^{\beta \log t}},$$
where $X\left(t\right)$ represents the percentage dissolved at a normalized time, t. Special attention should be given to the two coefficients: the first parameter, α, determines the proportion of the undissolved drug or molecule. This parameter is defined as a scale or position parameter [59,60,61]. On the other hand, β is a shape parameter and it determines the dissolution rate [59,60,61]. The Gompertz model is highly useful for determining the comparison between different in vitro drug release profiles, which, however, must exhibit good solubility and an intermediate release rate [61]. 

#### 3.2.2. Korsmeyer–Peppas Model

The KPM is a semi-empirical, comprehensive equation that simulates drug release from several delivery systems, primarily for polymeric systems. The KPM is expressed as follows [56,57,59,60,61,62,63]: (2)$$X\left(t\right)=k_{KP}t^{n}$$

In Equation (2), $k_{KP}$ represents the constant of proportionality (in s^−1^), and n is the release rate index as a drug release indicator of the mechanism. The constant rate and release exponent are known to depend on dosage form geometry, as well as on the dominating process (e.g., diffusion), but also on other factors governing diffusion and relaxation rates [61]. The KPM, if the polymer relaxation process is the slowest step [61], results in a zero-order drug release kinetics, so that n=1 [61]. Thus, the KPM can be suited for several MagS DD cases. 

#### 3.2.3. Fitting Quality

For the identification of the candidate equation most suited to the model, the MagS DD data trend was performed by comparing two figures of merit, namely the correlation coefficient ($R^{2}$), or so-called coefficient of determination, and the root mean square error (RMSE). The correlation coefficient is computed as [64]: (3)$$R^{2}=1-\frac{\sum_{n=1}^{N}{\left(y_{i}-{\hat{y}}_{n}\right)}^{2}}{\sum_{n=1}^{N}{\left(y_{i}-\bar{y}\right)}^{2}}\;.$$

In Equation (3), ${\hat{y}}_{1}$, ${\hat{y}}_{2},\ldots ,{\hat{y}}_{n}$ are the n-th predicted values, while $y_{1}$, $y_{2}$, … $y_{n}$ are the n-th observed values being $\bar{y}=\frac{1}{N}\overset{N}{\underset{n=1}{\sum }}y_{n}$. In Equation (3), the numerator is the sum of squares of errors generated by the model under consideration, while the denominator indicates the average of the sum of squares of errors generated by the reference model. Since $R^{2}\in \left[0,1\right]$, the best mathematical model will be selected for $R^{2}\to 1$, as it is the most suitable and confirms the drug release kinetics. 

Alongside the correlation coefficient, the RMSE is one of the most vital indicators for verifying the validity of a given mathematical model, as it measures the difference between the values predicted by the predictive model and the actual values [64]:(4)$$RMSE=\sqrt{\frac{1}{N}\overset{N}{\underset{n=1}{\sum }}{\left|y_{n}-\bar{y}\right|}^{2}}.$$

The variable retains its usual meaning. As previously mentioned, it provides an estimate of the accuracy of the predictive model: the lower the value of the RMSE, the better the model. Consequently, the model that produces a better approximation and representation of the starting data is characterized by having the lowest RMSE value.

Therefore, we based our study on the simultaneous estimation and evaluation of these two figures of merit, aiming to find the kinetic model that best approximated the data of MagSs for DD. The expression of the theoretical equation to describe the release kinetics has been plotted to derive the predicted data values and graph them together with the initial ones, with the subsequent calculation of the previously described error metrics. The validity of the mathematical model, where validity means the ability to approximate as much as possible the release kinetics of the drug contained by the scaffold under examination, has been evaluated. 

## 4. Results and Discussion

We selected three different types of MagSs [41,42,43,44,46,49,50,51] as cases of study to find a suitable DD model. These MagSs are interesting since they present different biomaterial-MNP combinations and different types of drug or biomolecule loadings that have been tested as potential candidates for the magnetic DD strategy for TE and/or CT, relying on different release mechanisms, under the application of static or alternate magnetic fields, with different intensities and frequencies. Therefore, given the limited availability of the experimental data for MagSs for DD, we will model their response and establish, for the first time, a quantitative basis for their design and use. By solving the various kinetic models, the fitting model parameters, along with the error metrics, were evaluated. The fitting model parameters were correlated and linked to MagS properties (i.e., the saturation magnetization, $M_{S}$, and volume fraction of MNPs, $\phi_{m}$), and to extrinsic magnetic DD parameters. The results have been interpreted and critically analyzed.

The data from the magnetic microsprouter from [41] (taken from Figure 4 pag. 4; Figure 6 pag. 6 from [41]), are presented in Figure 2. The figures of merit to evaluate the fitting quality and select the most suited kinetic model are provided in Table 2. The fitting results are presented in Figure 2. The best fitting model is the KPM model. In [41], methylene blue (MB) and DTX were considered as the drugs to be released. The cumulative release is studied in the case of a non-magnetic scenario and for a static MF applied to the MagSs. For the MB, few differences (~10%) are found between the two cases (17.81 μg vs. 19.97 μg) [41]. It must be reported that the authors tested different MF strengths, and a non-linear quadratic ($X\left(\left|B\right|\right)=18.6\;\left[\%/mT^{2}\right]\cdot x^{2}+3.76\left[\%/mT\right]\cdot x-1.7101\left[\%\right],\;R^{2}=0.98$) trend of the maximum-released drug concentration as a function of the strength of the magnetic flux density vector (B, in mT) can be derived from the data from [41]. The authors did not report the exact values of $M_{S}$ and $\phi_{m}$; therefore, it is not possible to directly relate the DD data to the material properties. However, the magnetic force exerted on the MagSs by the action of the external B field is $F\propto M_{S}\nabla B$ [65,66]; thus, by increasing the field strength, the gradient increases and so the force increases too. Therefore, if the DD mechanism is dictated by the mechanical deformation, a higher $M_{S}$ can ensure a faster and more sustained release. However, the drug molecule can affect the release kinetics and the MagS features can impact on its release too. The most relevant case is DTX release. From the fitting coefficients reported in Table 3, and the actual docetaxel release [41] (data from pag. 6, Figure 6 from [41]), whose profiles are presented in Figure 2b, it can be noticed that the magnetically triggered and controlled release results in a larger proportionality constant (~twofold) and in a super transport condition (n>0.89) [59,60,61]. From these quantitative findings, the boost of the release resulting from the MF action mediated by the magnetic biomaterial is evident. 

In [42], magnetic alginate scaffolds were considered. The data shown in Figure 3 are taken from Figure 6, pag. 43 from ref. [42]. A 40–60% difference can be observed in the cumulative release of BSA and DOX between the non-magnetic and magnetic cases. This difference can be explained by the fact that the mechanical deformation induced by the magnetic force exerted by the external MF causes a faster diffusion. To explain the drug release, for this MagS, the Gompertz model is most suited, as can be seen from the results reported in Table 4 (on average, $R^{2}=0.95$ vs. $R^{2}=0.82$, RMSE=0.03 vs. RMSE=0.05). From the fitting coefficients reported in Table 5, we can observed a ~70% lower fraction of undissolved drugs and ~2–3-fold-higher dissolution rates for the magnetically mediated drug release of BSA and DOX. 

These findings are partially corroborated by the KPM coefficients, since in Table 5, larger time constants were observed, in the presence of a quasi-Fickian release in magnetic cases. Therefore, the MF can act on the release kinetic, modifying its features.

The data from [43], Figure 3A, pag. 1398, for the poly(N-isopropylacrylamide) embedding 25% wt. of Fe_3_O_4_ MNPs developed for DD for TE and CT under dynamic MF excitation were considered and are reported in Figure 4. For this MagS, the release is triggered by an alternate-current (AC) magnetic-flux density field working at 220−260 kHz and with strengths of some mT [43,44]. The best model is the KPM, and its fitting coefficients are presented in Table 6. Few details are available for the magnetic properties of this MagS. However, in the presence of super transport, the AC MF (H, in A/m) causes heat dissipation for the MNPs embedded in the MagS. The magnetic energy is converted into power per unit volume according to the law $Q_{m}=\pi f\mu_{0}{\left|H\right|}^{2}\chi^{\prime \prime }$, $\chi^{\prime \prime }$ being the out-of-phase component of the complex magnetic susceptibility of the MNPs that ultimately depends on $\phi_{m}$ and $M_{s}$ [45]. The magnetic energy converted in heat lead for the possibility of breaking chemical bonds causes phase changes or, as in this case, increases the permeability of membranes due to the increase in the system’s temperature [43,44,45].

With this knowledge, the release from the MBG-PCL 3D-printed scaffolds loaded with Fe_3_O_4_ MNPs under dynamic MF exposure could be better interpreted [46]. Indeed, the magneto-thermal conversion triggers a high DOX release useful for CT applications. In [46], (see Figure 5, pag. 7950 from [46]), the experimentally measured curves of drug release over time present a general sigmoidal trend, as shown in Figure 5. In [46], different MagS compositions have been tested, so information about how $\phi_{m}$ and $M_{s}$ relate to kinetic parameters can be modeled for the first time. In Figure 5a, the data and the results from the fitting are shown. The best model fitting the release data for the bare MBG/PCL and the magnetic MBG/PCL scaffolds is the Gompertz model, as can be inferred from Table 7. By observing the retrieved coefficients for the Gompertz model (Table 8), a pattern can be identified. It can be noticed that a nonlinear relationship between the percentage of released drug at the final time (t=250 h) and $M_{s}$ can be identified (Figure 5b). It must be reported that the increase in the volumetric content of MNPs in the biomaterials leads to a slight modification of and increase in MagS porosity [41,42,43,44,45,46,47,48,49,50,51,52,53]. However, despite the porosity changes, we hypothesize that the differences in the DD mechanism are mediated by the interactions between the MF and the MagS. In other words, we assume that the released value is therefore a function of MagS features, considering that the MF parameters were fixed [49]. Therefore, α and β parameters, representing, respectively, the undissolved proportion and the dissolution rate, must be linked to MagS saturation magnetization. As it can be observed in Figure 5c, the α parameter is characterized by an approximately constant trend for all MagS compositions, hence being independent from the fraction of MNPs contained in the biomaterial ($\phi_{m}$). On the other hand, observing β in Figure 5c, it is possible to infer that the dissolution rate depends on $M_{s}$ in a linear way ($\beta =-0.015\cdot M_{s}-0.38,\;R^{2}=0.97$). 

These findings represent a relevant quantitative result of this work, as they allow us to understand how, through the MF-MagS interaction, the release kinetics can greatly improve. Moreover, the finding poses an interesting challenge to material science, i.e., the investigation of mathematical and physical phenomena that rule the interaction between MagSs and MFs. To further support the conclusions reported in the above discussion, another mesoporous calcium–iron-based MagS was studied in [50]. 

This chemically doped MagS releases gentamicin under the action of 1.47 kA/m, 232 kHz MF [50]. The release curves from Figure 11, pag. 1287 ref. [50] are reported in Figure 6a. It is possible to observe that the MagS can release ~10% more drugs after 40–60 h than its nonmagnetic counterpart. From Table 9, it can be seen that both the modified Gompertz model (Equation (1)) and the KPM are suited to model kinetics. The release constant for the two cases has a 6% difference, and the transport process is diffusion-dominated (n≤0.45), leading to similar undissolved proportions across the two cases, whilst a 13% difference in the value of the dissolution rate coefficient is obtained, as shown in Figure 6b. 

Finally, we focused on the investigation of a relevant case in which MagSs allowed drug transport by an external MF for triggering and controlling the release of very different drugs and biomolecules. In [49], nanoporous ferrogels were loaded with agents of three very different molecular weights and of diverse types [51] (see Figure 3, pag. 69 in [51]). Thus, the controlled release, mediated by the MF, has been evaluated, even under different drug functions or bioactive agent loadings. As first, the release of mitoxantrone for therapeutic purposes was performed in the presence and absence of MFs (Figure 7a). It can be noticed that, in the magnetic scenario, an increased release of ~40% is observed. The Gompertz model is the best fit (see Table 10). A faster release kinetic and dissolution rate (~3 time) are observed, as well as a modification of the release mechanism (i.e., from non-Fickian to super transport) for the two cases can be noticed (see Table 11). Then, the releases of plasmid DNA condensed with polyethylene diamine (with a molecular weight of ~106) and chemokine SDF-1α (with a molecular weight of ~8000) [51] from the MagS were modeled. It must be noticed that, in this case, the stimulation has been changed from 2 h intervals to 30 min, with magnetic stimulation achieved through 120 on/off cycles lasting 2 min, so that the highly macroporous structure was reversibly deformed, with subsequent release [51]. According to Figure 7b,c and the results from Table 10, the Gompertz equation faithfully simulates the envelope of the release kinetics of the loaded drug and of the other agents, with a high molecular weight. It is worth noting that the release of the chemokine is not sufficient and relatively low values are reached (Figure 7b). However, by relying on the findings shown in Figure 7c, considering the perspective of using MagSs as platforms for TE and CT, an ~8% release of plasmid DNA, in several hundreds of minutes, can be achieved from the MagSs using the external MF. The limited DNA release falls under a non-Fick mechanism that deserves further study, but could be optimized by relying on MagS properties and external MF parameters. 

For the sake of clarity, in Table 12, we provide a complete summary of the best models fitting the drug release data from works [41,43,44,46,49,50,51]. It can be observed that, generally, the KPM is the most suitable theoretical framework to interpret drug release from MagSs, either for a DD triggered by static or alternate magnetic fields. We can further notice that the MagS composition and manufacturing approach cannot be easily related to drug delivery performances and to the best model; therefore, future studies to elucidate this point are needed. 

## 5. Conclusions

This work dealt with the modeling of magnetic scaffolds that are magneto-responsive devices originating from the combination of traditional biomaterials and magnetic nanoparticles. MagSs can be used as platforms for magnetically triggered and controlled drug release for tissue engineering and cancer therapy. Therefore, static or dynamic external magnetic stimuli can be used to remotely control tissue repair or activate the release of drugs for tumor treatment. After having carefully analyzed the literature, we have identified that, despite several kinds of MagSs being manufactured, characterized, and tested, there is no quantitative framework to understand, interpret, or design magnetically triggered drug release. Furthermore, since most MagSs suffer from a local inhomogeneous distribution of drug concentration and burst release, models for their optimization are needed. By relying on experimental data from the literature, in this work, we proposed a modeling framework and a quantitative interpretation of different magnetic scaffolds. We found that, generally, the Gompertz model can better fit the drug release data, with a low error (RMSE<0.01, $R^{2}>0.9$), for MagSs triggered by static or dynamic magnetic fields. We found that the intrinsic magnetic properties of MagSs are key features for selecting the most suited and effective drug release mechanism, as well as to tune the kinetics of release. It was observed that the undissolved proportion and the dissolution rate decrease as MagS saturation magnetization increases, whilst the time constant decreases. These findings can be useful to material scientists to design innovative MagSs for DD. 

This work has the potential to be the quantitative basis for subsequent studies that aim at clarifying the physical phenomena and mechanisms underlying the interaction between magnetic fields and MagSs that determine the release of drugs or biomolecules for tissue engineering and/or cancer therapy. The proposed model can serve as a basis to design and plan experimental studies to further elucidate the mechanisms of DD mediated by MagSs. Therefore, future works must deal with the manufacturing, characterization, experimental tests, and modeling of MagSs for DD applications.

## Figures and Tables

**Figure 1 bioengineering-11-00573-f001:**
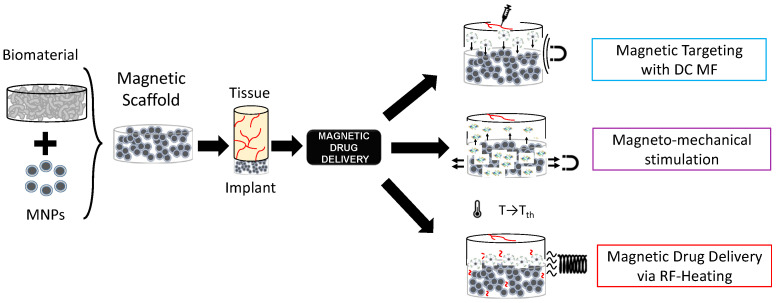
Graphical representation of the concept of magnetic scaffolds as the combination of magnetic nanoparticles and biomaterials, and their use for drug delivery applications.

**Figure 2 bioengineering-11-00573-f002:**
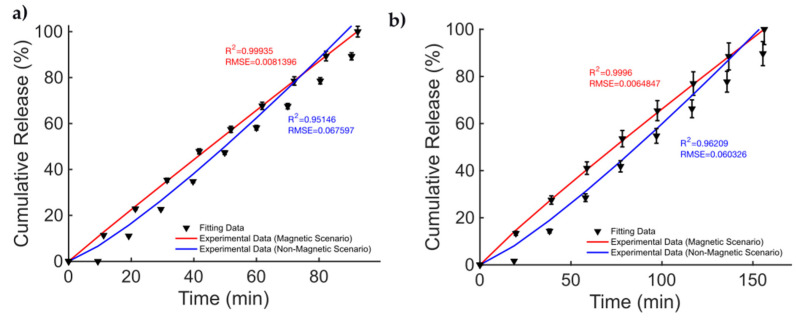
Release profile data for methylene blue in the cases of (**a**) non-magnetic and magnetic materials. Docetaxel release over time (**b**) for a non-magnetic membrane and in the presence of an external MF.

**Figure 3 bioengineering-11-00573-f003:**
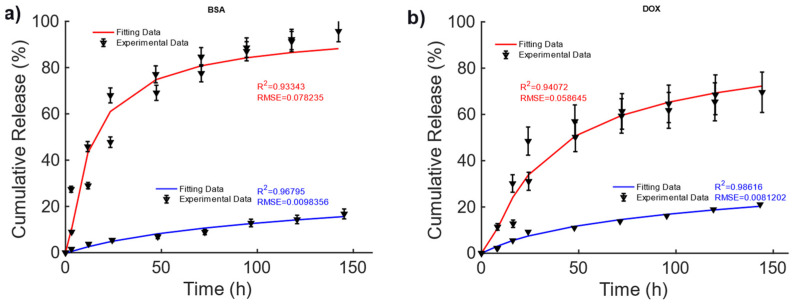
Cumulative release of (**a**) BSA and (**b**) DOX from hollow alginate non-magnetic (blue curve) and magnetic scaffolds (red) under the actions of a magnetic field being turned on and off.

**Figure 4 bioengineering-11-00573-f004:**
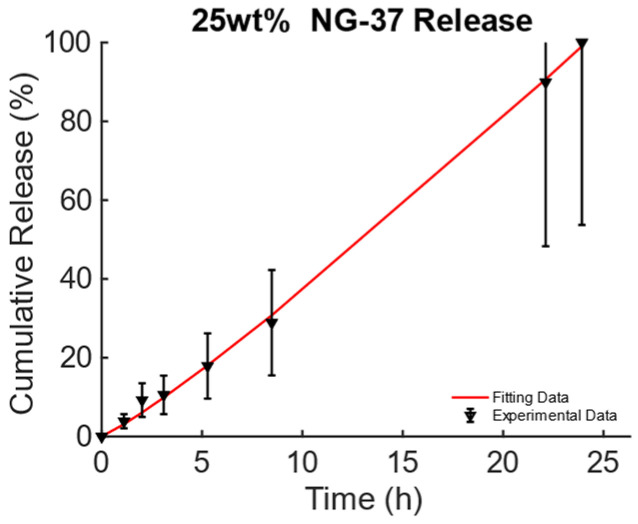
Cumulative release from magnetic poly(N-isopropylacrylamide) embedding 25% wt. of Fe_3_O_4_ MNPs triggered by magneto-thermal conversion using a dynamic MF.

**Figure 5 bioengineering-11-00573-f005:**
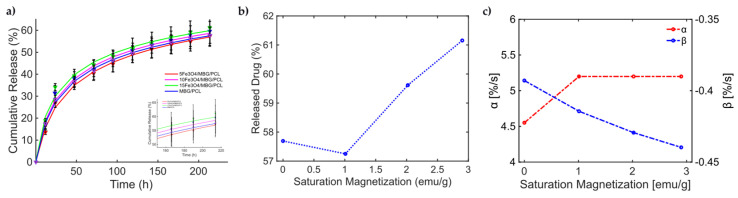
(**a**) Cumulative release from different MBG/PCL scaffolds with loadings from 5% to 15% of Fe_3_O_4_ MNPs [51]. (**b**) Released drug as a function of MagS saturation magnetization. (**c**) Variation in the undissolved proportion and the dissolution rate as a function of MagS saturation magnetization.

**Figure 6 bioengineering-11-00573-f006:**
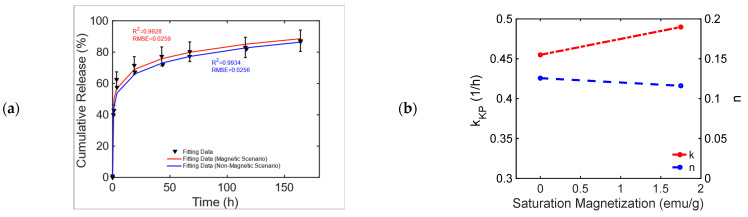
(**a**) Cumulative release from a bare and an iron-doped mesoporous bioglass triggered by an RF MF [50]. (**b**) Variation in the KPM parameters as a function of MagS saturation magnetization.

**Figure 7 bioengineering-11-00573-f007:**
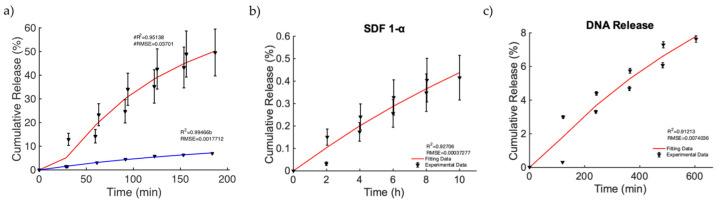
(**a**) Release of mitoxantrone for the non-magnetic (blue curve) and magnetic scenarios (red curve). (**b**) Release of chemokine SDF-1α from a ferrogel MagS. (**c**) Release of DNA material over time from the MagS.

**Table 1 bioengineering-11-00573-t001:** Literature analysis of MagSs for DD.

Work	Biomaterial	MNPs	Manufacturing	Mechanism	Drug/GFs	DD App.
[41]	PDMS	Ferromagnetic carbonyl iron	Incorporation	Static MF Mechanical deformation	MB, DTX	CT
[42]	Alginate	Fe_3_O_4_	Blending	Static MF Mechanical deformation	BSA, DOX	CT
[43,44]	poly(N-isopropylacrylamide)	Fe_3_O_4_	Dissolution Evaporation	Dynamic MF Magneto-thermal	-	TE, CT
[46]	MBG-PCL	Fe_3_O_4_	3D printing	Dynamic MF Magneto-thermal	DOX	TE, CT
[47]	Gelatin	Fe_3_O_4_	Blending	Static MF Mechanical deformation	-	TE
[48]	Hap	Fe_2_O_3_ Fe_3_O_4_	Chemical doping	Pulsed MF Mechanical deformation	DOX	CT
[50]	MBG	Fe_3_O_4_	Chemical doping	Dynamic MF Magneto-thermal	Gentamicin	TE
[51]	Alginate	Fe_3_O_4_	Blending	Dynamic MF	Mitoxantrone plasmid DNA chemokine	TE
[52]	Bioactive glass	Fe_3_O_4_	3D printing	Passive release	Mitomycin C	TE

**Table 2 bioengineering-11-00573-t002:** Comparison of fitting quality for the Gompertz and KP models for magnetic microsprouters.

	Gompertz Model		KPM	
	$R^{2}$	RMSE	$R^{2}$	RMSE
DTX (magnetic scenario)	0.9410	0.0888	0.9996	0.0069
DTX (non-magnetic scenario)	0.9636	0.0747	0.9934	0.0318
MB (magnetic scenario)	0.9423	0.0857	0.9994	0.0091
MB (non-magnetic scenario)	0.9631	0.0738	0.9931	0.0319

**Table 3 bioengineering-11-00573-t003:** Fitting coefficients for the two models for the drug released from magnetic microsprouters.

	Gompertz Model		KPM	
	α	β	$k_{KP}$ (1/min)	n
DTX (magnetic scenario)	624.75	−1.64	0.0093	0.9261
DTX (non-magnetic scenario)	4.6357 × 10^3^	−2.04	0.0024	1.1984
MB (magnetic scenario)	316.05	−1.67	0.0121	0.9758
MB (non-magnetic scenario)	1.71 × 10^3^	−2.07	0.0044	1.2102

**Table 4 bioengineering-11-00573-t004:** Comparison of fitting quality for the Gompertz and KP models for alginate SPIO scaffolds.

	Gompertz Model		KPM	
	$R^{2}$	RMSE	$R^{2}$	RMSE
BSA (magnetic scenario)	0.9334	0.0836	0.9398	0.0795
BSA (non-magnetic scenario)	0.9679	0.0112	0.9863	0.0073
DOX (magnetic scenario)	0.9407	0.0627	0.8780	0.0899
DOX (non-magnetic scenario)	0.9862	0.0092	0.9849	0.0096

**Table 5 bioengineering-11-00573-t005:** Fitting coefficients for the two models for the drug release alginate SPIO scaffolds.

	Gompertz Model		KPM	
	α	β	$k_{KP}$ (1/h)	n
BSA (magnetic scenario)	5.3917	−0.7583	0.1667	0.3641
BSA (non-magnetic scenario)	7.1249	−0.2701	0.0055	0.6796
DOX (magnetic scenario)	9.2091	−0.6727	0.0698	0.4827
BSA (non-magnetic scenario)	6.1927	−0.2733	0.0112	0.5897

**Table 6 bioengineering-11-00573-t006:** Comparison of fitting quality for the Gompertz and KP models for alginate SPIO scaffolds and associated derived coefficients.

Gompertz Model		KPM	
$R^{2}$	RMSE	$R^{2}$	RMSE
0.9532	0.0925	0.9985	0.0163
α	β	$k_{KP}\;$(1/h)	n
118.9692	−2.2373	0.0277	1.1266

**Table 7 bioengineering-11-00573-t007:** Comparison of fitting quality for the Gompertz and KP models for Fe_3_O_4_ MBG/PCL scaffolds.

	Gompertz Model		KPM	
	$R^{2}$	RMSE	$R^{2}$	RMSE
MBG/PCL	0.9986	0.0072	0.981	0.0265
5Fe_3_O_4_/MBG/PCL	0.9967	0.0112	0.9824	0.0257
10Fe_3_O_4_/MBG/PCL	0.9904	0.0183	0.9869	0.0225
15Fe_3_O_4_/MBG/PCL	0.9986	0.0273	0.9868	0.0228

**Table 8 bioengineering-11-00573-t008:** Fitting coefficients for the two models for the drug release for Fe_3_O_4_ MBG/PCL scaffolds.

	Gompertz Model		KPM	
	α	β	$k_{KP}$ (1/h)	n
MBG/PCL	4.551	−0.3929	0.0824	0.3709
5Fe_3_O_4_/MBG/PCL	5.1993	−0.4144	0.0769	0.3804
10Fe_3_O_4_/MBG/PCL	5.1993	−0.4294	0.0946	0.3474
15Fe_3_O_4_/MBG/PCL	5.1993	−0.4397	0.1079	0.3263

**Table 9 bioengineering-11-00573-t009:** Fitting results for the bare and iron-doped mesoporous bioglasses triggered by an RF MF.

	Gompertz Model		KPM	
	$R^{2}$	RMSE	$R^{2}$	RMSE
15Ca	0.9934	0.0256	0.983	0.041
10Fe5Ca	0.9928	0.0259	0.995	0.0216
	α	β	$k_{KP}$ (1/h)	n
15Ca	0.8318	−0.3276	0.4898	0.1161
10Fe5Ca	0.8313	−0.2834	0.4549	0.1257

**Table 10 bioengineering-11-00573-t010:** Comparison of fitting quality for the Gompertz and KP models for the active alginate MagS tested for the release of mitoxantrone, peptide, and DNA.

	Gompertz Model		KPM	
	$R^{2}$	RMSE	$R^{2}$	RMSE
Mitoxantrone (non-magnetic)	0.9944	0.0022	0.9882	0.0031
Mitoxantrone (magnetic)	0.9515	0.0405	0.9392	0.0454
SDF 1-α	0.927	0.0004	0.9101	0.0005
DNA	0.9121	0.0083	0.9066	0.0085

**Table 11 bioengineering-11-00573-t011:** Fitting coefficients for the Gompertz and KP models for the active alginate MagS tested for the release of mitoxantrone, peptide, and DNA.

	Gompertz Model		KPM	
	α	β	$k_{KP}$	n
Mitoxantrone (non-magnetic)	9.6482	−0.2491	0.0013 1/min	0.7664
Mitoxantrone (magnetic)	40.1393	−0.7774	0.0043 1/min	0.9231
SDF 1-α	7.6068	−0.1464	0.0008 1/h	0.7305
DNA	15.4473	0.2812	0.0003 1/min	0.8578

**Table 12 bioengineering-11-00573-t012:** Summary of the best model fitting the drug release data.

Ref.	[41]	[42]	[43,44]	[46]	[50]	[51]
Best model	KPM	Gompertz	KPM	Gompertz	Gompertz	Gompertz

## Data Availability

Data will be available upon request.

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
