# Peer review of "Modeling of Magnetic Scaffolds as Drug Delivery Platforms for Tissue Engineering and Cancer Therapy"

_bioengineering, 2024, doi:10.3390/bioengineering11060573_

Round 1
Reviewer 1 Report
Comments and Suggestions for Authors
The article adopts three different types of research as case studies to find suitable DD models. It models their reactions and establishes a quantitative basis for their design and use for the first time. By solving various kinetic models, the fitted model parameters and error metrics are evaluated. The overall writing quality of the manuscript is good, and the following modifications are suggested:
Q1: The "Introducing" section is too lengthy, suggesting condensation.
Q2: Enhance the "Related Works and Case Study" section with more current research and analysis of gaps to highlight the focus of this paper.
Q3: Revise unclear text in some figures and re-upload.
Q4: How are the effects of different preparation techniques on drug release systems considered in simulations?
Q5: In future research, it is recommended to supplement with actual experimental data and compare it with the simulation model.
Comments on the Quality of English LanguageModerate editing of English language required.
Author Response
Dear Reviewer,
the replies to your questions and doubts can be found in the attached file.
Best regards

Reviewer 2 Report
Comments and Suggestions for Authors
1. Small typo and grammatical mistakes are there. Please correct it.
2. The overall English and 'spell-checked' and 'grammar-checked' needs to improve.
Author Response

(The authors gave the same response as above.)

Reviewer 3 Report
Comments and Suggestions for Authors
Authors have presented a sound concept in the area of drug delivery. The literature search, content of the article and presentation is in line with a flow. Discussing about challenges of current MagS and proposing models/ strategies to overcome makes apt conclusion.
Author Response

(The authors gave the same response as above.)

Round 2
Reviewer 1 Report
Comments and Suggestions for Authors
The author has responded to the review comments. The revised manuscript can be accepted.
Comments on the Quality of English LanguageMinor editing of English language required
Reviewer 2 Report
Comments and Suggestions for Authors
I consider that the authors have made all the suggested changes, Now, I propose the paper for final publication.